# Multi-Scale Continuous Normalizing Flows

## Abstract

We introduce a multi-scale variant of Continuous Normalizing Flows, and explore the computation of likelihood values. We also introduce a Wavelet-version of the model. However, we find that this formulation is flawed in the computation of BPD, and explore ways to alleviate this problem.

## 1. Introduction

Reversible generative models derived through the use of the change of variables technique (Dinh et al., 2017; Kingma & Dhariwal, 2018; Ho et al., 2019; Yu et al., 2020) are growing in interest as alternatives to generative models based on Generative Adversarial Networks (GANs) (Goodfellow et al., 2016) and Variational Autoencoders (VAEs) (Kingma & Welling, 2013). A change of variables approach facilitates the transformation of a simple known probability distribution such as Gaussian noise into a more complex model distribution, such as images. Reversible generative models using this technique are attractive because they enable efficient density estimation, efficient sampling, and admit exact likelihoods to be computed. A promising variation of the change-of-variable approach is based on the use of a continuous time variant of normalizing flows (Chen et al., 2018; Grathwohl et al., 2019), which uses an integral over continuous time dynamics to transform a base distribution into the model distribution. This approach uses ordinary differential equations (ODEs) specified by a neural network, or Neural ODEs.

In this work, we consider a direct multi-resolution approach to continuous normalizing flows. While state-of-the art GANs and VAEs exploit the multi-resolution properties of images, and recently top performing methods also inject noise at each resolution (Brock et al., 2019; Shaham et al., 2019; Karras et al., 2020; Vahdat & Kautz, 2020), only recently have normalizing flows exploited the multi-resolution properties of images, using wavelets (Yu et al., 2020).

[1]Anonymous Institution, Anonymous City, Anonymous Region, Anonymous Country. Correspondence to: Anonymous Author <anon.email@domain.com>.

Preliminary work. Under review by INNF+ 2021. Do not distribute.

We use Continuous Normalizing Flows (CNF) in a multi-resolution fashion to generate an image at finer resolutions conditioned on the immediate coarser resolution image. A high-level view of our approach is shown in Figure 1.

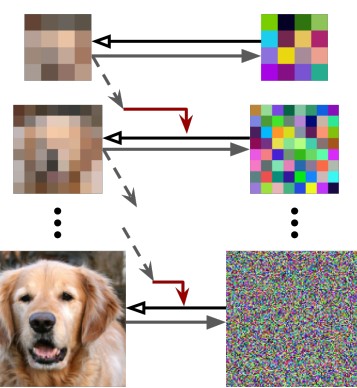

*Figure 1.* The architecture of our MSFlow-Image method. Continuous normalizing flows are used to generate images at each resolution, with finer resolutions being generated conditioned on the coarser image one level above.

## 2. Our method

Since images naturally exhibit structure in resolution, images can be decomposed into representations at multiple resolutions. We take advantage of this property by first decomposing an image in *resolution space* i.e. into a series of images at coarser resolutions : $(\mathbf{x}_0, \mathbf{x}_1, \ldots \mathbf{x}_S) = \mathbf{x}_{s \leq S}$. We then train an invertible generative model that normalizes this joint multi-resolution image into multi-resolution noise.

### 2.1. Normalizing Flows

We wish to train a generative model on a multi-resolution set of true images, i.e. find a probability distribution $p(\mathbf{x}_{s \leq S}) = p(\mathbf{x}_0, \mathbf{x}_1, \ldots, \mathbf{x}_S)$ that matches the true data distribution. Normalizing flows (Tabak & Turner, 2013; Jimenez Rezende & Mohamed, 2015; Dinh et al., 2017; Papamakarios et al., 2019; Kobyzev et al., 2020) are good candidates for such a model, as they are probabilistic generative models that perform exact likelihood estimates, and can be run in reverse to generate novel data from the model's distribution. This allows model comparison and measuring generalization to unseen data. Normalizing flows are trained

by maximizing the log likelihood of the input images. If a normalizing flow produces output $\mathbf{z}$ from an input image $\mathbf{x}$, the change-of-variables formula provides the likelihood of the image under this transformation as:

$$\log p(\mathbf{x}) = \log p(\mathbf{z}) + \log \left| \det \frac{d\mathbf{z}}{d\mathbf{x}} \right| \tag{1}$$

$\log p(\mathbf{z})$ is computed as the log probability of $\mathbf{z}$ under the noise distribution (typically standard Gaussian).

### 2.2. Joint multi-resolution representation

We depart from the typical usage of the multi-resolution representation of images by observing that had we not known that the images at different resolutions could be derived from one fine image, we now have a joint distribution over all possible images at all resolutions. Suppose $\mathbf{x}$ and $\mathbf{y}$ are two different images. Under this joint multi-resolution distribution, $(\mathbf{x}_{S-2}, \mathbf{x}_{S-1}, \mathbf{x}_S)$ and $(\mathbf{y}_{S-2}, \mathbf{y}_{S-1}, \mathbf{y}_S)$ are valid multi-resolution images, but so are $(\mathbf{y}_{S-2}, \mathbf{x}_{S-1}, \mathbf{y}_S)$ and $(\mathbf{x}_{S-2}, \mathbf{y}_{S-1}, \mathbf{x}_S)$. It so happens that our real data distribution of multi-resolution images are those multi-resolution data points that are correlated in resolution space. This is equivalent to the fact that among all possible single-resolution images, only those that have correlated pixels in width and height are real/natural images, as opposed to noise images without any correlation among pixels.

### 2.3. Multi-Resolution Normalizing Flows

We now wish to map the joint distribution of multi-resolution images $\mathbf{x}_{s \leq S}$ to "joint" multi-resolution noise $\mathbf{z}_{s \leq S}$. In this case, the multi-resolution change-of-variables formula is:

$$\log p(\mathbf{x}_{s \leq S}) = \log p(\mathbf{z}_{s \leq S}) + \log \left| \det \frac{\partial \mathbf{z}_{s \leq S}}{\partial \mathbf{x}_{s \leq S}} \right| \tag{2}$$

The multi-resolution structure of the data results in a simplification of the calculation of the Jacobian determinant. To illustrate this, choose a non-redundant basis of multi-resolution variables such that $\mathbf{z}_s$ at any resolution is linearly independent of $\mathbf{x}_{s+j}, j > 0$ at finer resolutions. This leads to the following block lower triangular structure in the variables:

$$
\begin{aligned}
&\log p(\mathbf{x}_{s \leq S}) \\
&= \sum_{s=0}^{S} \log p(\mathbf{z}_s) + \log \left| \det \begin{bmatrix} \frac{\partial \mathbf{z}_0}{\partial \mathbf{x}_0} & 0 & \cdots & 0 \\ \frac{\partial \mathbf{z}_1}{\partial \mathbf{x}_0} & \frac{\partial \mathbf{z}_1}{\partial \mathbf{x}_1} & \cdots & 0 \\ \vdots & \vdots & \ddots & \vdots \\ \frac{\partial \mathbf{z}_S}{\partial \mathbf{x}_0} & \frac{\partial \mathbf{z}_S}{\partial \mathbf{x}_1} & \cdots & \frac{\partial \mathbf{z}_S}{\partial \mathbf{x}_S} \end{bmatrix} \right| \\
&= \sum_{s=0}^{S} \left( \log p(\mathbf{z}_s) + \log \left| \det \frac{\partial \mathbf{z}_s}{\partial \mathbf{x}_s} \right| \right) \tag{3}
\end{aligned}
$$

We train a normalizing flow *at each resolution* to compute the likelihood of the image up to that resolution using Equation 3. This allows us to learn normalizing flows at each resolution independently, and in parallel.

Since the Jacobian determinant is a (block) lower triangular matrix, the non-zero off-diagonal elements don't contribute to the final log probability. Hence, we can freely condition each normalizing flow on the coarser images, by treating the coarser images as independent variables. This allows us to learn only the higher-level information at each resolution. We use this to our advantage, and train each normalizing flow $g_s$ between $\mathbf{x}_s$ and $\mathbf{z}_s$ conditioned on the immediate coarser $\mathbf{x}_{s-1}$ making a Markov assumption:

$$\mathbf{z}_0 = g_0(\mathbf{x}_0); \quad \mathbf{z}_s = g_s(\mathbf{x}_s \mid \mathbf{x}_{s-1}) \quad \forall s > 0 \tag{4}$$

Hence, equation 3 can be rewritten as:

$$
\begin{aligned}
\log p(\mathbf{x}_{s \leq S}) &= \log \mathcal{N}(g_0(\mathbf{x}_0); \mathbf{0}, \mathbf{I}) + \log \left| \det \frac{\partial g_0}{\partial \mathbf{x}_0} \right| \\
&+ \sum_{s=1}^{S} \left( \log \mathcal{N}(g_s(\mathbf{x}_s | \mathbf{x}_{s-1}); \mathbf{0}, \mathbf{I}) + \log \left| \det \frac{\partial g_s}{\partial \mathbf{x}_s} \right| \right) \tag{5}
\end{aligned}
$$

### 2.4. Multi-Resolution Continuous Normalizing Flows

We choose to use Continuous Normalizing Flows at each resolution (CNF) (Chen et al., 2018; Grathwohl et al., 2019), since they have recently been shown to effectively model image distributions using a fraction of the number of parameters typically used in normalizing flows (and non flow-based approaches). At each resolution, each CNF $g_s$ transforms its state (say $\mathbf{v}(t)$) using a Neural ODE (Chen et al., 2018) with neural network $f_s$:

$$\mathbf{v}(t_1) = g_s(\mathbf{v}(t_0) \mid \mathbf{c}) = \mathbf{v}(t_0) + \int_{t_0}^{t_1} f_s(\mathbf{v}(t), t, \mathbf{c}) \, dt \tag{6}$$

Chen et al. (2018); Grathwohl et al. (2019) proposed an instantaneous variant of the change-of-variables formula CNFs, which expresses the change in log-probability of the state of the Neural ODE i.e. $\Delta \log p_\mathbf{v}$ as a differential equation:

$$\Delta \log p_{\mathbf{v}(t_0) \to \mathbf{v}(t_1)} = - \int_{t_0}^{t_1} \mathrm{Tr} \left( \frac{\partial f_s}{\partial \mathbf{v}(t)} \right) dt \tag{7}$$

Hence, the ODE solver solves for the augmented state with the above differential, to obtain both the final state as well as the change in log probability simultaneously. Thus, the log probability at each resolution in eqs. (3) and (5) can be computed as:

$$\log p(\mathbf{x}_{s \leq S}) = \sum_{s=0}^{S} \left( \log p(\mathbf{z}_s) + \Delta \log p_{\mathbf{x}_s \to \mathbf{z}_s} \right) \tag{8}$$

We call this model MSFlow-Image.

In general, at each resolution (except the coarsest), the image $\mathbf{x}_s$ could first be converted to another representation $\mathbf{y}_s$ using a suitable orthogonal bijective transformation $T$ from $\mathbf{x}_s$ to $\mathbf{y}_s$ so that $\Delta \log p_{\mathbf{x}_s \to \mathbf{y}_s} = 0$:

$$\log p(\mathbf{x}_{s \leq S}) = \tag{9}$$
$$\sum_{s=0}^{S} \left( \log p(\mathbf{z}_s) + \Delta \log p_{\mathbf{y}_s \to \mathbf{z}_s} + \underbrace{\Delta \log p_{\mathbf{x}_s \to \mathbf{y}_s}}_{0} \right)$$

In the simplest case, $\mathbf{y}_s = \mathbf{x}_s$, which is MSFlow-Image. A more complex orthogonal transform to use is the Haar wavelet transform, we call this model **Multi-Scale Continuous Normalizing Flow - Wavelet (MSFlow-Wavelet)**. At each resolution, $\mathbf{x}_s$ is transformed into a composition of the 3 wavelet coefficients $\mathbf{w}_s$, and the coarser version $\mathbf{x}_{s-1}$, i.e. $\mathbf{y}_s = (\mathbf{w}_s, \mathbf{x}_{s-1})$. In this case, the conditioning becomes more obvious: each CNF maps the wavelet coefficients $\mathbf{w}_s$ to a noise sample $\mathbf{z}_s$ conditioned on $\mathbf{x}_{s-1}$ (see Figure 2), similar to WaveletFlow(Yu et al., 2020) which builds on Glow (Kingma & Dhariwal, 2018).

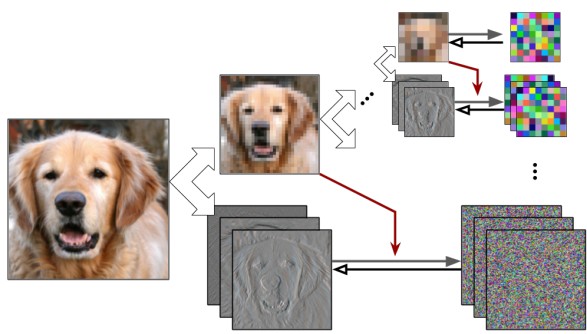

*Figure 2.* Architecture of MSFlow-Wavelet.

**Training**: The overall model is trained to maximize the log-probability of the joint multi-resolution image, given by Equation 8 as the sum of the likelihoods of the images at each resolution. Equivalently, our model is trained to minimize the Bits-per-dimension (BPD) of the image at finest resolution $S$ with $D_S$ pixels:

$$\text{bpd}(\mathbf{x}_{s \leq S}) = \frac{-\log p(\mathbf{x}_{s \leq S})}{D_S \log 2}$$
$$= \frac{-1}{D_S \log 2} \left[ \sum_{s=0}^{S} (\log p(\mathbf{z}_s) + \Delta \log p_{\mathbf{x}_s \to \mathbf{z}_s}) \right] \tag{10}$$

Since each CNF $g_s$ independently models the conditional distribution of the image at that resolution, we train each $g_s$ to minimize each $\text{bpd}(\mathbf{x}_{s' \leq s})$ step by step from the coarsest resolution ($s = 0$) to the finest resolution ($s = S$), having frozen $g_j : j \neq s$.

We use FFJORD (Grathwohl et al., 2019) as the baseline model for our CNFs. In addition, to speed up the training of FFJORD models by stabilizing the learnt dynamics, FFJORD RNODE (Finlay et al., 2020) introduced two regularization terms: the kinetic energy of the flow and the Jacobian norm. STEER (Ghosh et al., 2020) introduced temporal regularization by making the final time of integration stochastic.

**Generation**: Assuming each $g_s$ is invertible (which CNFs are), we may then generate images using ancestral sampling: we first sample $\mathbf{z}_s$'s from a latent noise distribution, and transform them backwards into image space progressively from coarser to finer resolutions through the CNFs:

$$\begin{cases} \mathbf{x}_0 = g_0^{-1}(\mathbf{z}_0) \\ \mathbf{x}_s = T^{-1}(\mathbf{y}_s) = T^{-1}(g_s^{-1}(\mathbf{z}_s \mid \mathbf{x}_{s-1})) & \forall \, s > 0 \end{cases} \tag{11}$$

## 3. Related work

Several prior works on normalizing flows (Kingma & Dhariwal, 2018; Song et al., 2019; Ma et al., 2019; Yu et al., 2020) build on RealNVP (Dinh et al., 2017). Although they achieve great results in terms of BPD and image quality, they nonetheless report results from significantly higher parameters and several GPU hours for training.

Our MSFlow-Wavelet model is quite similar to the recently published WaveletFlow(Yu et al., 2020). However, WaveletFlow builds on the Glow (Kingma & Dhariwal, 2018) architecture, while ours builds on CNFs (Grathwohl et al., 2019; Finlay et al., 2020). Moreover, WaveletFlow applies certain techniques to obtain better samples from its model. We have so far not used such techniques for generation, but they can potentially help generate better samples from our models.

## 4. Experimental results

We train MSFlow-Image and MSFlow-Wavelet models on the CIFAR10 (Krizhevsky et al., 2009) dataset at finest resolution of 32x32, and the ImageNet (Deng et al., 2009) dataset at 32x32, 64x64, 128x128. We build on top of the code provided in (Finlay et al., 2020)[1]. In all cases, we train using *only one* NVIDIA V100 GPU with 16GB.

**Ablation study on regularizers**: We perform an ablation study using the two regularizations mentioned above: with/without RNODE (Finlay et al., 2020), with/without STEER (Ghosh et al., 2020). We find that consistently in all cases, FFJORD RNODE achieves superior BPD in lesser time. In some cases, FFJORD fails to train.

**Ablation study on resolutions**: We train models with varying number of total resolutions. Increasing the number of

---

[1]https://github.com/cfinlay/ffjord-rnode

*Table 1.* Unconditional image generation metrics (lower is better in all cases): number of parameters in the model, bits-per-dimension, time (in hours). Most previous models use multiple GPUs for training, all our models were trained on only *one* NVIDIA V100 GPU. ‡As reported in (Ghosh et al., 2020). *FFJORD RNODE (Finlay et al., 2020) used 4 GPUs to train on ImageNet64. 'x': Fails to train. Blank spaces indicate unreported values.

| | CIFAR10 | | | IMAGENET32 | | | IMAGENET64 | | |
|---|---|---|---|---|---|---|---|---|---|
| | PARAM | BPD | TIME | PARAM | BPD | TIME | PARAM | BPD | TIME |
| **1-scale Continuous Normalizing Flow** | | | | | | | | | |
| FFJORD (Grathwohl et al., 2019) | | 3.40 | >5days | | 3.96‡ | >5days‡ | | x | x |
| FFJORD RNODE (Finlay et al., 2020) | | 3.38 | 31.84 | | 2.36‡ | 30.1‡ | | 3.83* | 64.1* |
| FFJORD + STEER (Ghosh et al., 2020) | 1.36M | 3.40 | 86.34 | 2.00M | 3.84 | >5days | 2.00M | | |
| FFJORD RNODE + STEER (Ghosh et al., 2020) | | 3.397 | 22.24 | | 2.35 | 24.9 | | | |
| **(OURS) 2-scale MSFlow-Image** | | | | | | | | | |
| 2-scale FFJORD | | 1.85 | 17.89 | | x | x | | x | x |
| 2-scale FFJORD RNODE | | 1.69 | 16.37 | | 1.92 | 26.20 | | 1.54 | 51.21 |
| 2-scale FFJORD + STEER | 0.48M | 2.04 | 18.76 | 0.16M | 2.36 | 20.12 | 0.16M | x | x |
| 2-scale FFJORD RNODE + STEER | | 1.74 | 18.43 | | 1.97 | 65.16 | | 1.58 | 66.76 |
| **(OURS) 3-scale MSFlow-Image** | | | | | | | | | |
| 3-scale FFJORD | | 1.54 | 21.51 | | 2.00 | 30.54 | | x | x |
| 3-scale FFJORD RNODE | | 1.32 | 21.48 | | 1.66 | 41.17 | | 1.21 | 60.89 |
| 3-scale FFJORD + STEER | 0.48M | 1.72 | 21.09 | 0.13M | 2.21 | 21.36 | 0.13M | x | x |
| 3-scale FFJORD RNODE + STEER | | 1.44 | 23.44 | | 1.68 | 54.05 | | 1.26 | 59.14 |
| **(OURS) 4-scale MSFlow-Image** | | | | | | | | | |
| 4-scale FFJORD | | 1.42 | 19.95 | | 1.84 | 30.63 | | x | x |
| 4-scale FFJORD RNODE | | 1.28 | 19.08 | | 1.62 | 42.60 | | 1.18 | 65.6 |
| 4-scale FFJORD + STEER | 0.64M | 1.88 | 17.73 | 0.17M | x | x | 0.17M | x | x |
| 4-scale FFJORD RNODE + STEER | | 1.44 | 17.60 | | 1.63 | 62.82 | | 1.36 | 66.2 |
| **(OURS) 5-scale MSFlow-Image** | 0.81M | 1.28 | 19.42 | | | | 0.22M | 1.17 | 71.33 |
| **(OURS) 6-scale MSFlow-Image** | 0.97M | 1.24 | 20.52 | | | | | | |
| **(OURS) 2-scale MSFlow-Wavelet** | 0.50M | 3.56 | 17.17 | 0.33M | 3.92 | 15.30 | | | |
| **(OURS) 3-scale MSFlow-Wavelet** | 0.76M | 3.69 | 13.99 | 0.51M | 4.00 | 17.70 | 0.51M | 4.04 | 37.82 |
| **(OURS) 4-scale MSFlow-Wavelet** | 1.03M | 3.77 | 13.94 | 0.69M | 4.02 | 16.83 | | | |
| **(OURS) 5-scale MSFlow-Wavelet** | 1.29M | 3.87 | 10.73 | | | | | | |

total resolutions consistently improves BPD across models with the same number of parameters per resolution, except in the case of MSFlow-Wavelet where we see the opposite case.

**Progressive training**: Since each resolution can be trained independently, we can train an MSFlow-Image model on ImageNet128 by training only the finest resolution (128x128) conditioned on 64x64 images for 1 epoch, and then attach that to a 4-resolution model trained on ImageNet64 from scratch. This 5-resolution ImageNet128 model gives a BPD of 1.13.

## 5. Fundamental flaw

However, we note that there is a fundamental flow to this calculation of BPD : we calculated the BPD of $\mathbf{x}_{s \leq S}$, while prior works report the BPD of $\mathbf{x}_S$. This implies that our model maps the joint distribution of images to joint noise, meaning our model includes images whose coarser versions do not correspond to the finest image. This does not apply to our MSFlow-Wavelet models since the wavelet formulation ensures the consistency of coarser images with respect to the fine image.

Hence, to find the likelihood of $\mathbf{x}_S$ under our MSFlow-Image model, the likelihood of $\mathbf{x}_{s \leq S}$ needs to be marginalized over the entire subspace of lower resolution images. This is intractable. To make it tractable, we could approximate this marginal using Monte Carlo integration, by sampling multiple lower-reolution images and summing over the respective joint likelihoods. Inevitably, this leads to much greated BPD values than the ones reported in Table 1. Hence, Table 1 is **not a fair comparison** to make, except for the MSFlow-Wavelet rows.

## 6. Conclusion

We have presented a Multi-Resolution approach to Continuous Normalizing Flows, and performed exact likelihood calculations on several benchmark datasets of images by training on a single GPU in lesser time with a fraction of the number of parameters of other competitive models. However, we found that our formulation is fundamentally flawed in the computation of BPD for a single image. We explored ideas over how to fix this issue. We found that formulations similar to the Wavelet formulation which ensure the consistency of coarsened images with respect to the finest image can help alleviate this problem.

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
