# OpenReview forum: "Multi-Scale Continuous Normalizing Flows"
_ICML.cc/2021/Workshop/INNF — Reject_

### Official Review · Reviewer_X4oh · 2021-06-10

**Rating:** Borderline Reject
**Confidence:** 3

**Summary:**

Authors introduce a new class of normalizing flows for modelling images. In particular they propose an architecture based on a multi-scale representation of the observations. One variant additionally relies on the family of orthogonal Haar wavelet transformation. They assess the modelling capacity of their models on standard image datasets.

**Justification For Rating:**

# Strengths
The introduced methodology is well motivated, as different resolutions of an image are strongly correlated, which have been empirically proven by methods like wavelets.
Normalizing flows are still lagging behind GANs in terms capacitiy to model complex distributions such as images. The proposed method is thus of relevance to the community.
The strength of the method is that the multi-resolution structure results in a block-diagonal Jacobian making this method tractable. Additionally, the parallel computation of each term of equation 9 makes the method scalable (although at sampling time it is linear in the depth of the resolutions).

# Weaknesses
I identify three limitations:
1/ One limitation that is stressed by authors themselves in Section 5, is that the joint distribution p(x_{s<=S}) is modelled, instead of the density of the image of interest p(X_S). There seems to be no simple way to marginalize the joint density. This objective may not be an issue itself for training purposes (is it a lower bound?), but is limiting for evaluation purposes.
2/ In terms of novelty, this work bears high similarity with Wavelet Flow (Yu et al., 2020), which also proposed such a multi-resolution architecture along with the same use of Haar wavelets. The only difference seems to be the choice of NFs: CNF vs RealNVP. Albeit powerful, CNF are known to be computationally expensive to train. Is there a specific reason why they should be better adapted for modelling images than affine-coupling methods (e.g. Real-VNP)?
3/ Empirically, the proposed wavelet variant, seems to be outperformed by a simple CNF (see Table 1). How is that so? Especially, the higher the resolution depth, the worse the performance gets. It appears to be faster though.

# Correctness / Clarity / Relation to prior work
It would be obviously quite relevant to compare to the RealNVP, Glow and Wavelet Flow models.
It would be useful to stress and explain the reason why MSFlow-Image models p(x_{s<=S}) whereas MSFlow-Wavelet models p(x_S). The paper is pretty well written otherwise.

---

### Official Review · Reviewer_5f5C · 2021-06-11

**Rating:** Borderline Accept
**Confidence:** 3

**Summary:**

The work proposes a multi-scale normalizing flow model. The model is over *joint* distribution of images at multiple resolutions x = (x_{low_res}, ..., x_{high_res}), where x_i, x_j are not necessarily correspond to the same images. The model has a conditioning mechanism to link images at different resolutions but optimize the joint likelihood. The model allows for independent training that accelerates training time.

Overall: The paper is clearly written, however, the results are not that consistent ...

1) Table 1 and Section 5 are confusing. Marginalization is a fair and correct point, to the best of my understanding. But why report incomparable metrics in Table 1 in the first place?

2) MC Approximate BPD is not reported, why is that? Is there any chance for reasonable estimation in high dimensional space? Maybe independent flows of different resolutions are the correct baseline to compare BPD? also classification accuracy score (https://arxiv.org/abs/1905.10887) might be useful to compare performance but need conditional versions.

3) Visual comparison is interesting to see. Especially given that BPDs are incomparable in most cases.

4) Is Wavelet flow is natural baseline here? How important are CNFs over GLOW that is one of the main selling points (line 137, col 2)? Is it mostly a matter of a number of parameters?

5) It is also interesting to see "FFJORD RNODE + STEER" training for fewer interactions in order to match training time.

Formatting:
- Consider Figure 1 [h!] -> [t!]
- Consider adding labels ys, xs, etc. to Figure 2

**Justification For Rating:**

The paper is well written. Direction is interesting.
Results are not so consistent but they can be fixed.

---

### Decision · Program_Chairs · 2021-06-15

**Decision:**

Reject

**Comment:**

We have decided to reject this paper due to the high similarity with another submission in this workshop by the same set of authors. In both papers the authors present work on continuous normalizing flows with multiple resolutions, and both papers propose a model where multiple resolutions are produced with wavelet transforms. As stated in this submission, the model with wavelet transforms is the only model in this paper that can be compared to other methods in terms of log likelihood, and that comparison with the other (non wavelet transform method) is unfair. The not-flawed method appears to be essentially the same as that of the other submission. Moreover, there is significant overlap in the text of the two submissions. Examples include the related work and the first sentence of the introduction. We cannot accept papers with such a large overlap to the workshop.